# Downregulation of sCD40 and sCTLA4 in Recovered COVID-19 Patients with Comorbidities

**DOI:** 10.3390/pathogens11101128

**Published:** 2022-09-30

**Authors:** Gwendolyne Méndez-Frausto, Susana Godina-González, César E. Rivas-Santiago, Edna Nungaray-Anguiano, Gretel Mendoza-Almanza, Bruno Rivas-Santiago, Carlos E. Galván-Tejada, Irma E. Gonzalez-Curiel

**Affiliations:** 1Laboratorio de Inmunotoxicología y Terapéutica Experimental, Unidad Académica de Ciencias Químicas, Universidad Autónoma de Zacatecas, Zacatecas 98160, Mexico; 2Laboratorio de Biomarcadores, Unidad Académica de Ciencias Químicas, Universidad Autónoma de Zacatecas, Zacatecas 98160, Mexico; 3CONACYT-Academic Unit of Chemical Sciences, Autonomous University of Zacatecas, Zacatecas 98160, Mexico; 4Unidad de Investigación Biomédica de Zacatecas, IMSS, Zacatecas 98000, Mexico; 5Unidad Académica de Ingeniería Eléctrica, Universidad Autónoma de Zacatecas, Zacatecas 98000, Mexico

**Keywords:** COVID-19, sCD40, sCTLA-4, diabetes mellitus, hypertension

## Abstract

The aim of this study was to analyze molecules associated with regulatory immune response in unvaccinated, recovered COVID-19 patients with and without diabetes mellitus (DM) and hypertension (HTN). We determined anti-SARS-CoV-2 nucleocapsid IgG in plasma by electrochemiluminescence immunoassay. The levels of sCD40, TGF-ß, IL-10, and sCTLA-4 were assessed by ELISA in the serum of the subjects, as well as in healthy donors. We observed that only half of the subjects in the non-comorbid group produced antibodies, whereas all subjects in comorbid groups were IgG-positive for the anti-SARS-CoV-2 nucleocapsid. High levels of sCTL-4 were observed in the non-comorbid group, and the level of IL-10 was observed to increase in seropositive subjects without comorbidities. TGF-ß concentration was similar in all groups studied. Finally, sCD40 decreased in the comorbid group. In conclusion, our results suggest that comorbidities such as DM and HTN alter the production of co-stimulatory inhibitory molecules sCTLA-4 and sCD40 in subjects recovering from mild COVID-19. The alterations observed here were independent of seropositivity, suggesting an effective humoral immune response against COVID-19 separate from the levels of co-stimulatory inhibitory molecules.

## 1. Introduction

Coronavirus disease (COVID-19) is caused by the novel human pathogen severe acute respiratory syndrome coronavirus (SARS-CoV)-2 [1]. The World Health Organization (WHO) considers COVID-19 a worldwide health problem due to its widespread global morbidity and mortality [2]. One of the main complications caused by COVID-19 is a hyperinflammatory syndrome associated with poor outcomes in severe COVID-19, which leads to acute respiratory failure and multiple damaged organs [3]. However, there is evidence that susceptibility to COVID-19-associated hyperinflammatory syndrome varies among individuals [4]. Moreover, comorbidities such as *diabetes mellitus* (DM), systemic arterial hypertension (HTN), and heart affection reflect poor outcomes in cases of severe COVID-19 [5,6].

Recent studies on the different stages of disease among COVID-19 patients observed distinct cytokine profiles between severe and non-severe COVID-19 cases [7,8]. In addition, lymphopenia in severe COVID-19 patients has been detected [9]. Such data suggest an immune response of hyperactivation and dysregulation against SARS-CoV-2. This abnormal response could contribute to the pathogenicity of COVID-19 [10].

Regulatory T cells (Tregs) are crucial in controlling inflammation and are fundamental to adaptive immunity [11]. Tregs express the cytotoxic T lymphocyte antigen 4 (CTLA-4) co-inhibitory receptor and produce immunosuppressive cytokines, such as TGF-ß and IL-10 [12]. However, there are inconsistent and contradictory results regarding the function of Tregs in SARS-CoV-2 infection [13]. Previous studies reported high levels of Tregs in severe and non-severe cases of COVID-19, suggesting possible immunosuppression and a possible decrease in virus clearance contributing to the pathological course [14,15]. In contrast, other authors demonstrated low levels of Tregs in severe SARS-CoV-2-infected patients, suggesting decreased inhibitory co-stimulation activity, producing immune hyperactivation and subsequent tissue damage [9]. Thus, the function of Tregs in SARS-CoV-2 patients is still unclear and needs further study [16,17].

Humoral immunity plays a fundamental role in the viral clearance of SARS-CoV-2. Specific antibodies are capable of neutralizing the virus for at least 3 months in recovered individuals after mild COVID-19 [18]. However, it has been observed that 1–10% of individuals who recovered from SARS-CoV-2 were seronegative [19]. In this sense, cellular response seems able to resolve the infection when humoral responses are not sufficient.

Immune alterations observed during COVID-19 might be associated with the deregulation of Tregs secretory molecules, worsening in comorbid patients. In this study, we evaluated the concentration of regulatory cytokines sCTLA-4, IL-10, and TGF-ß, as well as the molecule of co-stimulation of antigen-presenting cells, soluble CD40 (sCD40), in serum of recovered COVID-19 patients with and without comorbidities from the state of Zacatecas, Mexico.

## 2. Material and Methods

This work was approved by the Research Committee (COFEPRIS 19-CI-32-056-045) and by the Research Ethics Committee (CONBIOÉTICA-32-CEI-20180927), attaining the CI-UAE-08-2020 registry and 060-CEI-UAE-UAZ/2020, respectively. Informed consent was obtained from all participants according to the Declaration of Helsinki. All the proceedings were performed in the Laboratory of Immunotoxicology and Biomarkers of the Autonomous University of Zacatecas in Mexico. All donors were recruited from different health institutions throughout the state of Zacatecas. Study subjects were recovered COVID-19 patients previously diagnosed by PCR test and by clinical symptoms. Healthy controls did not present clinical COVID-19 symptomatology, and they were negative for SARS-CoV-2 antibodies. After the first contact and oral explanation of the experimental protocol, all the donors signed an informed consent agreement.

### 2.1. SARS-CoV-2-Specific IgG Levels

The samples were collected from May 2020 to August 2020 from various municipalities of the state of Zacatecas in Mexico. Blood samples from recovered patients of mild COVID-19 treated as outpatients were collected through venipuncture in a vacuum collection tube with EDTA-K2. Collected samples were centrifuged at 1500 rpm for 5 min to obtain plasma to determine N-specific IgG by ELISA according to the manufacturer’s protocol.

The seroprevalence was estimated based on the IgG isotype, since IgM does not provide long-term memory information [20]. In this sense, the detection of IgG makes it possible to evaluate the scope of herd immunity as a monitoring and epidemiological surveillance strategy. A blood sample was used from each participant with prior informed consent to determine the presence of anti-SARS-CoV-2 IgG antibodies by means of an electrochemiluminescence immunoassay, Elecsys^®^ Anti-SARS-CoV-2, Precicontrol, using a COBAS e411 autoanalyzer (Roche Diagnostics). Anti-N IgG titers were considered positive with a cut-off index of >1.

### 2.2. Subjects

A sociodemographic questionnaire was used to measure anthropometric variables (i.e., body mass, height, blood pressure, waist/hip ratio) of every patient. All study subjects provided signed written consent before any study interaction. We collected 78 serum samples from recovered COVID-19 patients through venipuncture. Samples were divided into non-comorbid seropositive (NC IgG+; n = 20), non-comorbid seronegative (recovered subjects who had not produced antibodies, NC IgG−; n = 20), and comorbid seropositive (diabetes (DM; n = 8) and hypertension (HTN; n = 10)). Serum samples from healthy donors with no SARS-CoV-2 infection precedent, no household contact with COVID-19 patients, and no comorbidities were collected as the control group (HD; n = 20).

### 2.3. ELISA

The levels of soluble CD40 (sCD40), TGF-ß, IL-10 (RayBiotech, Peachtree Corners, GA, USA), and soluble CTLA-4 (sCTLA-4) (Thermo Fisher Scientific, Waltham, MA, USA) were assessed by enzyme-linked immunosorbent assay (ELISA) in the serum of comorbid and non-comorbid COVID-19-recovered subjects, as well as healthy donors. Briefly, the assay sensitivity for CTLA-4 was settled at 0.13 ng/mL (sCTLA-4; REF: BMS276), TGF-ß at 18 pg/mL (cat. ELH-TGF-ß1), IL-10 at 1 pg/mL (cat. ELH-IL-10), and sCD40 at 50 pg/mL (cat. ELH-CD40). All procedures were carried out according to the manufacturer′s instructions. The signal was measured on a Thermo Scientific Multiskan FC photometer (v1.00.94; Thermo Scientific, Waltham, MA, USA).

### 2.4. Statistical Analysis

All data are presented as means ± standard deviation (SD) or median + range. One-way analysis of variance (ANOVA) or Kruskal–Wallis tests were applied depending on data normality. Subsequently, to test the differences among groups, we performed Dunn′s post hoc test or the Dunnett T3 test using Graph Pad Prism 8.0 (Graph Pad Software, San Diego, CA, USA). Finally, correlation analysis among inhibitory molecules and anthropometric data was carried out using R software v3.6.1 (RStudio, PBC, Boston, MA, USA). The level of statistical significance was set at 95% (*p* < 0.05).

## 3. Results

### 3.1. Titers of SARS-CoV-2 N-Specific IgG

The long-lasting immune response to COVID-19 was determined by assessing the titers of the SARS-CoV-2 anti-nucleocapsid (Anti-N) IgG antibodies in a cohort of 78 subjects. The seroprevalence was estimated based on the IgG isotype since IgM does not provide long-term memory information. 

Subjects were classified as healthy donors (HD, n = 20), recovered non-comorbid subjects (NC, n = 40), and recovered comorbid subjects (C, n = 18). More detailed information on the groups is shown in Table 1. All subjects in the comorbid group (10 with hypertension (HTN) and 8 with diabetes mellitus (DM)) were seropositive, and the non-comorbid group had 10 seropositive and 10 seronegative subjects. All healthy donors were seronegative.

### 3.2. Levels of Co-Inhibitory Molecules in Serum of COVID-19-Recovered Subjects

The serum levels of cytokines and molecules involved in the regulation and co-stimulation of the immune response, such as IL-10, TGF-ß, and sCTLA-4, were assessed by ELISA in healthy donors (HD IgG−), non-comorbid seronegative and seropositive subjects (IgG− and IgG+, respectively), and comorbid seropositive subjects (DM IgG+ and HTN IgG+). 

Levels of sCTLA-4 were higher in non-comorbid subjects than in healthy controls and comorbid donors (Figure 1A). Interestingly, no differences were observed after a stratified analysis between non-comorbid IgG+ and non-comorbid IgG− groups. In addition, there were no differences between DM IgG+ and HTN IgG+. However, among all studied groups, the highest expression of this molecule was observed in the non-comorbid IgG+ group (Figure 1B). 

The IL-10 levels did not show differences between control, non-comorbid, and comorbid groups (Figure 1C). However, after a stratified analysis, IL-10 levels were significantly higher in the non-comorbid IgG+ group in comparison to the non-comorbid IgG− group (Figure 1D).

We also analyzed the serum levels of TGF-ß, and no differences were observed between the non-comorbid and comorbid groups in comparison to the control group (Figure 1E,F).

Finally, we assessed the serum levels of the co-stimulatory molecule sCD40. We observed lower levels of sCD40 in the comorbid group than in the control group and the non-comorbid group (Figure 2A). Likewise, with the stratified analysis, we detected a significant reduction in the serum sCD40 levels of DM IgG+ and HAS IgG+ subjects compared to the control group (Figure 2B).

### 3.3. Principal Component Analysis and Correlation Analysis

We performed principal component analysis (PCA) to assess the relationship between humoral immunity and regulatory T response. Remarkably, the first and second principal components explain 37% and 13.4% of the variance, respectively. Indeed, samples from patients with DM IgG+ clustered more closely with HTN IgG+, while those of NC IgG+ and NC IgG− groups scattered away in the PCA (Figure 3A). Anthropometric variables such as BMI, WC, weight, and TGF-ß serum levels mainly explain the variability between our groups (Figure 3B,C).

A Pearson correlation matrix was performed. A low significant negative correlation was observed between the systemic blood pressure (SBP) and negative IgG values for the SARS-CoV-2 nucleocapsid, and a significant negative correlation was observed between CTLA-4 and IL-10 in the control group (r = −0.39, *p* = 0.045) (Figure 4A). In the non-comorbid group, a significant negative correlation was observed between TGF-ß and sCD40 (r = −0.22, *p* = 0.048), IL-10, and height (r = −0.21, *p* = 0.008) (Figure 4B). For the comorbid group (Figure 4C), a significant negative correlation was observed between sCD40 and IgG-N (r = −0.66, *p* = 0.0001), weight (r = −0.46, *p* = 0.0066), BMI (r = −0.37, *p* = 0.0076), WC (r = −0.40, *p* = 0.0008), and WHR (r = −0.40, *p* = 0.0011). In contrast, TGF-ß was positively correlated with IL-10 (r = 0.43, *p* = 0.0213).

## 4. Discussion

One of the aims of this study was to assess the seropositivity to IgG against N-protein of SARS-CoV-2 in a cohort of recovered COVID-19 patients with and without comorbidities. We observed that only half of the subjects in the non-comorbid group produced antibodies, whereas all subjects in comorbid groups were IgG-positive to the anti-SARS-CoV-2 nucleocapsid. Similarly, Macedo et al. detected high seropositivity in all comorbid groups analyzed, where DM and HTN were the comorbidities with the highest incidence [21].

We also evaluated CTLA-4, an important co-inhibitory molecule produced by Tregs, in the serum of non-comorbid and comorbid patients who recovered from mild COVID-19. We observed high levels of CTLA-4 in the non-comorbid group. Interestingly, the highest levels were found in the seropositive non-comorbid group, suggesting a greater cell inhibitory response to avoid an exacerbated immune response during the recovery period. Regarding the comorbid group, a recent study showed that CTLA-4 gene expression is shared in five different comorbid groups during COVID-19, including DM and hypertension [22]. Intriguingly, CTLA-4 levels in seropositive comorbid groups did not increase, suggesting that these subjects neutralized and cleared the virus more effectively, rendering the production of CTLA-4 unnecessary to evade an exacerbated immune response. Contrary to our results, other studies have reported high amounts of sCTLA-4 in TCD4+ lymphocytes of SARS-CoV-2 seropositive subjects older than 50 years of age [23]. However, as shown in Figure 4, we did not observe a correlation with age in any of our studied groups. Therefore, the discrepancy observed could be related to the severity of the disease and not to the age of the studied subjects. For example, Schub et al. observed that the expression of CTLA-4 in subjects with severe COVID-19 was higher than in those with mild COVID-19 [24].

The IL-10 levels showed no statistically significant differences between healthy controls, non-comorbid subjects, and comorbid subjects. However, after a stratified analysis, higher levels of IL-10 were found in the non-comorbid seropositive group than in the seronegative groups. In addition, although CTLA-4 and IL-10 did not correlate, the highest concentration of both molecules was observed in the non-comorbid seropositive group. This could be related to the observation that the soluble isoform of CTLA-4 induces the secretion of IL-10 [25].

Regarding TGF-β levels, there were no significant changes in serum concentrations in any of the studied groups (Figure 1E,F). Similar findings were reported elsewhere in patients infected with SARS-CoV-2 [26,27]. In addition, previous research has reported high levels of TGF-ß during the first 3–7 days of infection, after which they decrease [28]. These results suggest that during COVID-19 recovery, TGF-β levels are normal. Following this observation, Che et al. proposed TGF-β blockers as a possible treatment against severe COVID-19, reporting that the low concentration of TGF-ß induced an easy recovery, reducing the probability of pulmonary edema and pulmonary fibrosis [29].

Interestingly, no differences were found in the TGF-ß levels of hypertensive subjects (HTN IgG+), contrary to previous studies that observed higher levels of TGF-ß in HTN subjects than in subjects without HTN, suggesting a possible mechanism for the management of hypertension [30]. The above findings suggest that TGF-ß expression during recovery is more similar to COVID-19 than underlying comorbidity.

The co-stimulatory molecule CD40 is present in B cells and antigen-presenting cells (APCs), and the presence of this molecule is critical for the development of humoral and cellular immunity. Therefore, CD40 is mainly necessary for differentiation, maturation, and immunoglobulin class switching in B cells.

A decrease in CD40 levels was observed in the comorbid groups (DM and HTN) (Figure 2A,B) despite the presence of anti-nucleocapsid SARS-CoV-2 in comorbid groups. These data suggest that in the comorbid groups, a humoral immune response was effective during mild COVID-19. Previous studies have focused on the importance of co-stimulatory molecules in the maintenance and function of Tregs [31]. In this sense, CD40 suppression does not induce Treg development, and CD40 blockers suppress Tregs in humans [32,33]. 

Conversely, CD40-activated B cells (CD40-B) stimulate the generation of Tregs by CD4+ T cells, and the inhibitory effect of the Tregs is dependent on CTLA-4 [34].

Our findings showed that the variables studied are closely related to the anthropometric data evaluated, specifically WC, BMI, weight, and WHR (Figure 3). Especially influential is sCD40, which negatively correlated with weight, BMI, and WHR in the comorbid group (Figure 4C). Similar results were observed in an obese mouse model, in which CD40 depletion increased the inflammation in adipose tissue, increasing body weight and contributing to the development of metabolic syndrome [35]. There are discrepancies in the literature regarding the impact of DM and HTN on sCD40 and sCTLA-4. Some authors have shown that both molecules are increased in comorbidities [36,37], while others found low levels of these molecules [38,39]. These suggest that the production of sCD40 and sCTLA-4 is related to the clinical state of the disease and glycemic control and depends on complications associated with the comorbidity, such as cardiovascular alterations, among others [40,41,42]. In this sense, the effects of DM and HTN on the molecules mentioned above are not yet clear. However, our study provides information on the behavior of sCD40 and CTLA-4 in subjects with comorbidities who recovered from mild COVID-19. Controversially, CD40 has been positively correlated with clinical parameters of obesity and associated with protection against obesity [43]. Thus, more studies are necessary to clarify if obesity interferes with the expression of these co-stimulatory molecules in COVID-19.

It has been reported that obesity is related to DM and HTN [44,45].

## 5. Conclusions

The results of our study suggest that comorbidities such as diabetes mellitus and hypertension alter the production of co-stimulatory inhibitory molecules sCTLA-4 and sCD40 in subjects recovering from mild COVID-19. The alterations observed here were independent of seropositivity, suggesting an effective humoral immune response against COVID-19 separate from the levels of co-stimulatory inhibitory molecules. However, the low response of the regulatory molecules could be due to a decrease in co-stimulatory molecules. This could explain why patients with comorbidities are at higher risk of complications or have longer periods of symptoms during recovery.

## Figures and Tables

**Figure 1 pathogens-11-01128-f001:**
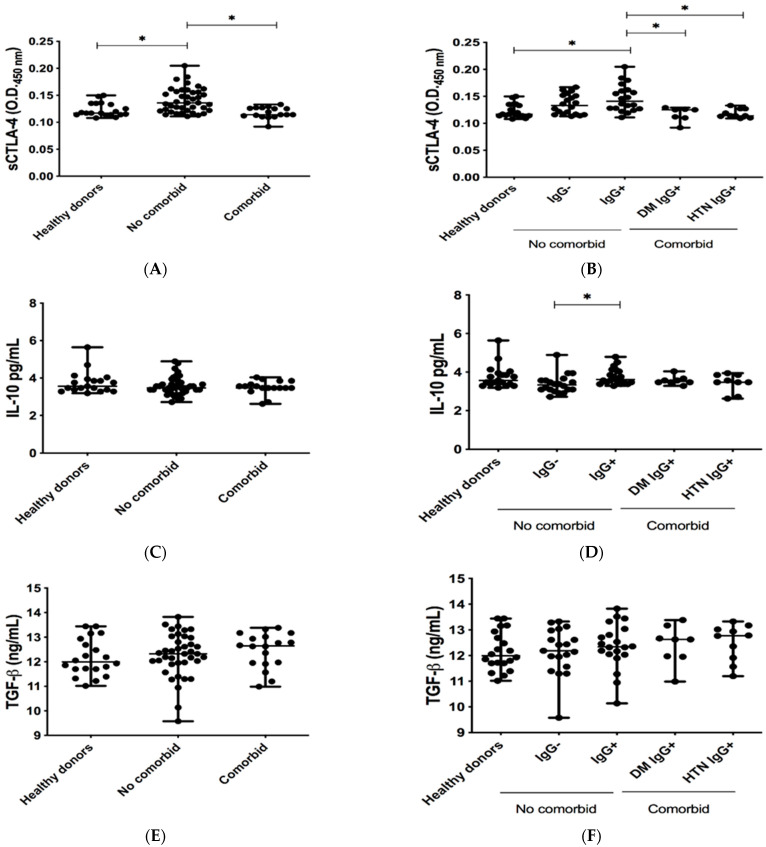
Determination of co-stimulation inhibitory molecules in patients recovered from COVID-19. (**A**,**B**) Optical density values at 450 nm (OD450) of CTLA-4, (**C**,**D**) concentration levels of IL-10, and (**E**,**F**) concentration levels of TGF-ß assessed by ELISA in serum of healthy donors, recovered non-comorbid COVID-19 subjects, and recovered comorbid COVID-19 subjects. The Kruskal–Wallis test and Dunn′s post hoc test were used for statistical analysis. Data are presented as median + range. * *p* < 0.05 was considered significant.

**Figure 2 pathogens-11-01128-f002:**
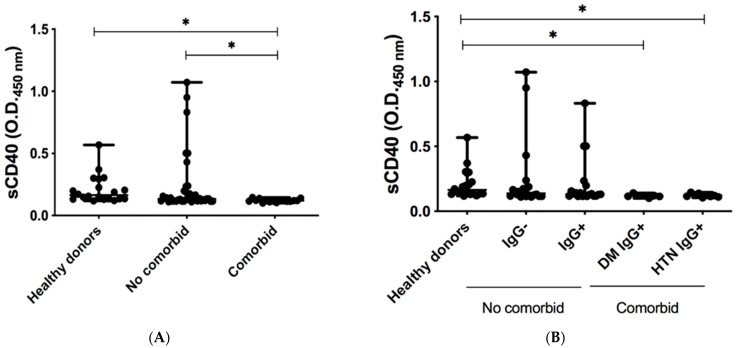
Determination of the co-stimulation molecule sCD40 in patients recovered from COVID-19. (**A**) Optical density values at 450 nm (OD450) of sCLA-4 in serum of healthy donors, recovered non-comorbid COVID-19 subjects, and recovered comorbid COVID-19 subjects. (**B**) Optical density values at 450 nm (OD450) of sCLA-4 in serum of healthy donors, recovered non-comorbid seronegative (IgG−) and seropositive (IgG+) subjects, and recovered comorbid seropositive (DM IgG+ and HTN IgG+) subjects by ELISA. The Kruskal–Wallis test and Dunn′s post hoc test were performed for statistical analysis. Data are presented as median + range. * *p* < 0.05 was considered significant.

**Figure 3 pathogens-11-01128-f003:**
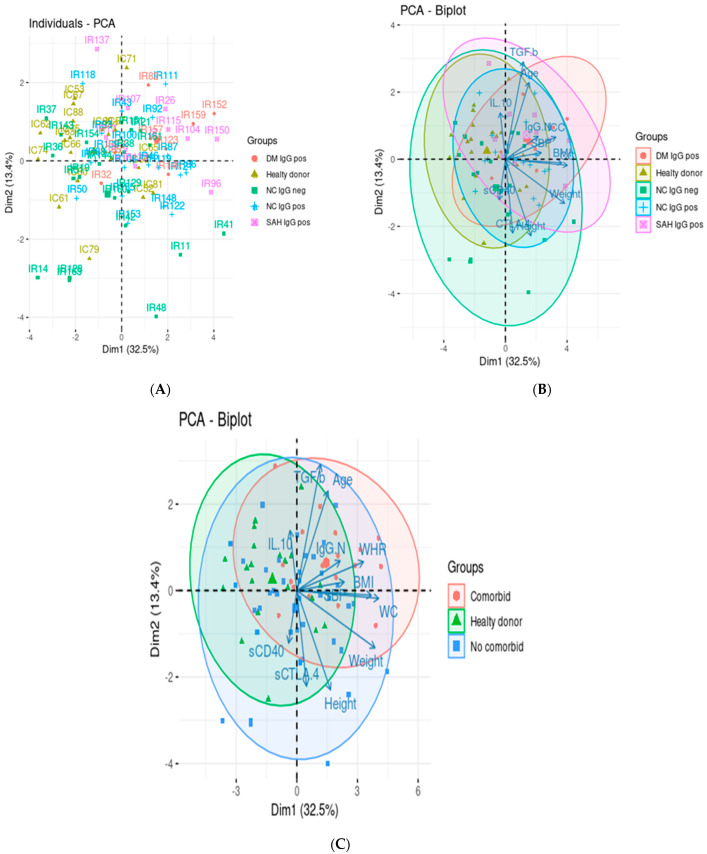
Principal component analysis. The score plot (**A**) of PCA showing the pattern of the individuals stratified as healthy donors (n = 20), seropositive non-comorbid subjects (n = 20), seronegative non-comorbid subjects (n = 20), comorbid diabetes mellitus subjects (DM; n = 8), and comorbid hypertension subjects (HTN; n = 10). (**B**) Bi-plot showing the relationships between the variables studied and anthropometric data stratified in the five groups. (**C**) Bi-plot showing the relationships between the variables studied and anthropometric data stratified in three groups, namely healthy donors, non-comorbid subjects, and comorbid subjects. The numbers are our internal identification subject. X- and Y-axes contain the values of the first and second principal components produced by the PCA.

**Figure 4 pathogens-11-01128-f004:**
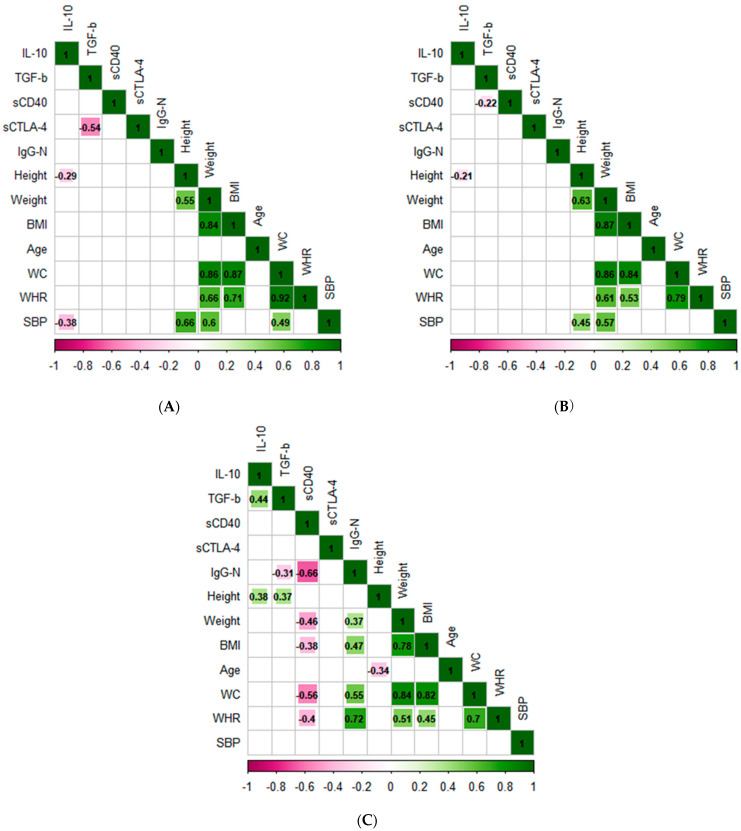
Correlation between co-stimulation inhibitory molecules and anthropometric data in recovered COVID-19 subjects. (**A**) Healthy donors; (**B**) non-comorbid group; (**C**) comorbid group. The numbers inside the cells correspond to the value of the correlation, the squares indicate correlations with statistically significant differences (*p* < 0.05), and the correlations that did not show significant differences remain blank. The horizontal color bar indicates the direction of the correlation values (pink is negative, green is positive).

**Table 1 pathogens-11-01128-t001:** Characteristics of the subjects included in this study.

Characteristics of Subjects	Healthy Donors (HD) n = 20	Non-Comorbid (NC) n = 40	Comorbid (C) n = 18	Reference Range
Gender (F/M)	13/7	23/17	10/8	
Age (y)	37.40 ± 9.87 ^$^	37.08 ± 10.43 ^$^	51.50 ± 10.63 ^$,^*^,+^	
**Comorbid (n)**				
*Diabetes mellitus* (DM)	-	-	8	
Hypertension (HTN)	-	-	10	
**Anthropometric Data**				
Height (m)	1.60 + 0.38 ^#^	1.64 + 0.39 ^#^	1.66 + 0.38 ^#^	
Weight (kg)	62.25 + 52.30 ^#^	73.05 + 73.20 ^#^	84.80 + 83.62 ^#,^*	
BMI (kg/m^2^)	24.98 + 24.67 ^#^	26.56 + 21.70 ^#^	30.01 + 22.02 ^#,^*	
WC (cm)	80.50 + 39.0 ^#^	92.0 + 74.0 ^#^	97.0 + 41.0 ^#,^*	
WHR	0.84 ± 0.09 ^$^	0.87 ± 0.09 ^$^	0.91 ± 0.07 ^$,^*	
SBP (mmHg)	117.0 + 22.0 ^#^	98.50 + 104.0 ^#,^*	130.0 + 50.0 ^#,+^	
**Antibody Test Results**				
IgG-N ^Cut-of index^	0.066 + 0.12 ^#^	5.29 + 153.4 ^#^	62.81 + 151.5 ^#,^*^,+^	<1.0

Female/male (F/M), body mass index (BMI), waist circumference (WC), waist/hip circumference ratio (WHR), systemic blood pressure (SBP). Data are presented as ^#^ median + range, ^$^ mean ± SD. * *p* < 0.05 vs. healthy donors, ^+^
*p* < 0.05 vs. non-comorbid group.

## Data Availability

Data that support the findings of this study are available within the article, and from the corresponding author upon request.

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
