# Peer review of "Downregulation of sCD40 and sCTLA4 in Recovered COVID-19 Patients with Comorbidities"

_pathogens, 2022, doi:10.3390/pathogens11101128_

Round 1

Reviewer 1 Report

In the study, the authors investigated the costimulatory inhibitory molecules (sCTLA-4, TGF-beta, sCD40, IL-10) in healthy donors (HD), and COVID-19 patients with or without comorbidities (diabetes and hypertension). They claimed that comorbidities alter the production of those inhibitory molecules.

There are many problems in the study. The major problems:

1. The results do not support the conclusion. In two out of four tested inhibitory molecules (IL-10 & TGF-beta), the comorbid group did not show a difference compared to either the no comorbid group or healthy donors group (Figure 1C & 1E).

2. The authors used the Turkey Post-Hoc method to test for statistical significance. However, Tukey post-hoc test should be used when the sample sizes for each group are equal. The participants in each group were different (40 vs 18 vs 20) so the statistical method was incorrectly used.

3. The sample size was too small. For instance, the patients in the diabetes group were 8, and the patients in the hypertension group were 10.

4. The sample variation was too high. In table 1, the IgG-N in the No-comorbid group, the variation of 51.16 is much greater than the mean of 36.50.

Some minor problems:

5. Table 1 layout was wrong in that the patient numbers in each group were mismatched. The healthy donors' group should have 20 participants but in the table, it was shown n=18.

6. It was said the anti-SARS-CoV-2-COVID19-IgG antibodies were tested (lines 93, 98, 104, and so on). They could examine the anti-SARS-CoV-2 antibody but there is no “anti-SARS-CoV-2-COVID19-IgG” antibody.

Even though the authors divided each of the “no comorbid” and “comorbid” groups into two groups and got some significance (the clinical meaning is unclear), there is no consistent pattern in the four tested molecules. I am not persuaded by the presented data.

Author Response

Dear Reviewer:

We appreciate your dedicated time to carefully and thoroughly review our work.

Please find below a point-by-point description of all the changes and suggestions made to the manuscript. The changes made by us are highlighted in the manuscript using yellow shading. Please see the attachment.

Kind regards,
Irma G. Curiel

Reviewer 2 Report

Major:

- Was the blood collection time point standardize with respect to onset of first symptoms or end of symptoms? Can a variable time of collection impact/introduce biais in the outcome of the study?

- Subjects: Why did the authors only collected blood samples from patients that had presented mild symptoms? Were any of the subject presenting both DM and  HTN?

- Controls: Why did the authors choose healthy patients with no conditions as control? DM and  HTN might be associated with cytokines/chemokines on their own. For instance, multiples previously published articles mentioned association of sCD40 with cardiovascular disease or CTLA-4 and diabetes. Authors need to thoroughly address their choice of controls with references to support their decision. It is especially important because some data only show significance with the wrong comparison: In Fig 2B, DM/IgG+ and HTN/IgG+ are not significantly different from No comorbid IgG+ patients for sCD40, there are only different from healthy controls that do not account for DM and HTN.

- Is the data relative to Tcell count/Treg count available from those patients? Analysis/clustering with respect to that might provide insight in the comparisons the authors are trying to make.

- Similarly, analysis/clustering with respect to Anti-SARS-CoV2-COVID19-IgG patients level might provide insight in the comparisons the authors are trying to make.

Minor:

- Line 71: subject missing in the sentence (However, XXX have been).

-Line 100-101: Add reference to statement.

- Figure resolution appears low, authors should introduce higher quality figures to their manuscript.

Author Response

(The authors gave the same response as above.)

Reviewer 3 Report

Mendez-Frausto et al analysed the levels of co-stimulatory and regulatory molecules, soluble CD40 (sCD40), TGF-ß, IL-10 and soluble CTLA-4 (sCTLA-4) by ELISA in recovered COVID-19 individuals with comorbidities (diabetes mellitus and hypertension) compared to individuals without any comorbidity. The authors found that the group with comorbidities had lower levels of sCD40 and sCTLA-4 in serum compared to the groups without comorbidity. However, this alteration was independent of SARS-CoV-2 seropositivity.

It is crucial to understand the factors which may impact on immune responses to SARS-CoV-2 in individuals with specific comorbidities. The topic of this manuscript is relevant. Unfortunately, this paper does not link the levels of regulatory molecules to protection or the risk to be infected. Please find below some comments to improve the current manuscript.

Major comments:

1) Lines 62-64: this sentence is unclear. Do the authors mean less Tregs in severe cases or the opposite ? Paragraph about Tregs must be clarified

2) Line 65: The authors say “SARS-CoV-2 infected” patients. Do they mean severe or non-severe cases ?

3) What is the difference between the healthy donor group and no comorbid seronegative group ? Is it only a difference in household contacts with COVID-19 patients ? It should be clarified in the manuscript.

4) Do the authors have previous data from SARS-CoV-2 seronegative individuals with DM or hypertension ? Did they also observe a downregulation sCD40 and sCTLA4 ? The comparison seropositive versus seronegative patients with comorbidities could help to understand if the basal level of these molecules is low or if the low level is linked to the recovery phase post-infection.  

5) Line 98: is it plasma ? Based on the methods, it looks it is serum. Please clarify.

6) Figure 4B is not cited in the text.

Minor comments:

1) Line 51: reflect without s ?

2) Line 56: I would suggest to delete “However”

3) Line 71: typo “it has been observed that”

4) Line 71: I would suggest “1-10% of SARS-CoV-2 recovered individuals were seronegative”

5) Line 89: “by clinical operational definition”. Do the authors mean symptoms ?

6) Line 90: I would add “healthy” before controls

7) Line 93: I would suggest to write “SARS-CoV-2 specific IgG levels” or “SARS-CoV-2 specific IgG responses”

8) Line 98: I would suggest to write “N-specific IgG” to be more accurate.

9) Line 136: typo “Titers”. I would suggest “Titers of SARS-CoV-2 N-specific IgG”

10) Lines 139-141: sentence already written in Methods. Please delete the repeat.

11) Line 142: cut-off information should be moved to the Methods.

12) Line 143: remove the semi-colon and add a dot.

13) Line 200: typo “post”

14) Line 340: do the authors mean the highest concentration of IL-10 ? Please clarify.

15) Line 345: “it has been”

16) Lines 373 and 377 are nearly similar. Please delete the repeat.

Author Response

(The authors gave the same response as above.)

Reviewer 4 Report

In this manuscript, Méndez-Frausto et al. measured proteins associated with regulation of the immune response, including soluble CTLA-4, IL-10, TGFB, and soluble CD40 in patients who have recovered from COVID-19. sCTLA-4 was found to be higher in recovered COVID-19 patients with no comorbidities compared to patients with comorbidities and healthy controls. Specifically, IgG+ seroconverted patients had greater sCTLA-4 compared to healthy controls, IgG+ DM patients, and IgG+ HTN patients. Differences in IL-10 expression were only seen between the IgG+ and IgG- non-comorbid groups (higher in IgG+). No significant differences were seen in TGFB expression. Lower sCD40 expression was seen in the comorbid group compared to the non-comorbid group and healthy controls, and DM IgG+ and HTN IgG+ patients had lower sCD40 compared to healthy controls. Furthermore, sCD40 expression was negatively correlated with weight, BMI, and WHR in the comorbid group. Given the greater risk of disease severity in DM and HTN patients, it is prudent to understand how the immune response to COVID-19 differs in these patients and the work presented here is of interest to the field. However, I have a few questions and concerns that should be addressed before publication is considered:

1)      The manuscript could use some careful grammatical editing.

2)      The lack of individuals with comorbidities in the healthy control group makes it a bit difficult to assess whether the differences seen are due to the diabetes or hypertension or are perhaps just due to differences in the severity of COVID symptoms, as these individuals are at higher risk of severe COVID. Are the authors able to include individuals with diabetes and hypertension without COVID infection in their analysis? If not, is there anything known about how diabetes or hypertension impact the expression of CTLA-4 and CD40 that the authors could include in their discussion?

3)      The rate of seroconversion seems quite low in the non-comorbid group. Did the authors find this surprising? Or is this typical of mild cases?

4)      How long after COVID infection were the samples obtained? This could also impact the levels of any immune-related markers.

Author Response

(The authors gave the same response as above.)

Round 2

Reviewer 1 Report

The authors have improved the manuscript even though some major problems cannot be fixed, such as the low sample size and huge variation. 

Reviewer 2 Report

Authors made good efforts to address all reviewers comments.

Reviewer 4 Report

I appreciate the authors' responses to my comments. The article is now suitable for publication.